# Double-stranded RNA prevents and cures infection by rust fungi

Rebecca M. Degnan [1]✉, Louise S. Shuey[2], Julian Radford-Smith[3], Donald M. Gardiner [4], Bernard J. Carroll [1], Neena Mitter [4], Alistair R. McTaggart[4,5] & Anne Sawyer [1,4,5]✉

Fungal pathogens that impact perennial plants or natural ecosystems require management strategies beyond fungicides and breeding for resistance. Rust fungi, some of the most economically and environmentally important plant pathogens, have shown amenability to double-stranded RNA (dsRNA) mediated control. To date, dsRNA treatments have been applied prior to infection or together with the inoculum. Here we show that a dsRNA spray can effectively prevent and cure infection by *Austropuccinia psidii* (cause of myrtle rust) at different stages of the disease cycle. Significant reductions in disease coverage were observed in plants treated with dsRNA targeting essential fungal genes 48 h pre-infection through to 14 days post-infection. For curative treatments, improvements in plant health and photosynthetic capacity were seen 2–6 weeks post-infection. Two-photon microscopy suggests inhibitory activity of dsRNA on intercellular hyphae or haustoria. Our results show that dsRNA acts both preventively and curatively against myrtle rust disease, with treated plants recovering from severe infection. These findings have immediate potential in the management of the more than 10-year epidemic of myrtle rust in Australia.

[1] School of Chemistry and Molecular Biosciences, The University of Queensland, St Lucia, Queensland, Australia. [2] Department of Agriculture and Fisheries, Queensland Government, Ecosciences Precinct, Dutton Park, Queensland, Australia. [3] School of the Environment, The University of Queensland, St Lucia, Queensland, Australia. [4] Queensland Alliance for Agriculture and Food Innovation, Centre for Horticultural Science, The University of Queensland, St Lucia, Queensland, Australia. [5] These authors contributed equally: Alistair R. McTaggart, Anne Sawyer. ✉email: r.degnan@uq.edu.au; a.sawyer@uq.edu.au

Rust fungi (Pucciniales) are obligate plant pathogens with widespread impacts across agriculture, forestry, and natural ecosystems[1]. Yield losses of up to 70% are regularly documented in globally important calorie crops such as wheat and soybean[2], and invasive rust fungi have caused epidemics in natural ecosystems leading to population decline and localised extinctions of endemic species[3–5].

Management of diseases caused by rust fungi relies on synthetic fungicides[6] or breeding and deploying resistant genotypes[7–10]. These approaches are largely confined to cropping systems as both fungicides and resistance breeding have limitations in non-agricultural contexts. Fungicides can be toxic to other organisms in the environment and run-off into waterways[11], while breeding for resistance is also restrained in perennial crops and other perennial species, and is further limited by rapidly evolving pathogens that overcome natural or bred resistance[12,13]. Due to the long-lived nature of perennial tree species, a curative treatment option that is effective at any time in the infection cycle will be crucial to manage disease.

Rust fungi of perennial trees in native ecosystems include *Austropuccinia psidii*, the cause of myrtle rust, and species of *Cronartium*, such as *C. ribicola*, the cause of white pine blister rust. Myrtle rust impacts the Myrtaceae family, including genera such as *Eucalyptus*, *Psidium*, and *Syzygium*. Its invasion in Australia has led to a predicted extinction event of 16 rainforest tree species in a single generation, many of which have significant environmental or cultural significance[14]. In Australia, the fungicide triadimenol has been sprayed on highly-susceptible hosts of *A. psidii* in an attempt to manage myrtle rust. While the treatment was initially successful, myrtle rust symptoms returned following completion of the spray-regime[15]. White pine blister rust and fusiform rust (caused by *Cronartium quercuum* f. sp. *fusiforme*) significantly impact North American pines, leading to ecological cascades, estimated losses of more than $140 million USD annually[4,16], and the listing of whitebark pines under the Endangered Species Act. Management of rust fungi on perennial trees, in both the forestry industry and in natural ecosystems, is a pressing challenge.

In response to the bottlenecks arising from traditional pathogen management, new strategies include mixed-cultivar cropping[17,18] and resistance gene pyramiding[19] to slow pathogen evolution, and biological control using existing pathogen antagonists[20,21]. Amongst the most promising of these new approaches are double-stranded RNA (dsRNA) sprays, utilising the RNA interference (RNAi) mechanism, due to their environmentally-friendly, non-genetically modified (GM) nature[22].

RNA interference (RNAi) is a post-transcriptional gene silencing mechanism, triggered on recognition of dsRNAs to silence homologous messenger RNA (mRNA)[23–25]. RNAi machinery is largely conserved throughout eukaryotes, including across the Pucciniales[26]. RNAi has shown success as a treatment against several species of rust fungi, both through exogenous application and transgenic approaches[26–33]. Exogenous tests of RNAi in rust fungi have applied dsRNA together with the inoculum (co-inoculation), or prior to infection (preventative RNAi). Indeed, we previously showed that dsRNA targeting essential fungal genes can inhibit urediniospore germination and the development of appressoria on leaf surfaces when co-applied with urediniospores[26]. Curative RNAi (application of dsRNA following infection) has not yet been explored in fungal plant pathogens, with just one report in the oomycete *Plasmopara viticola* (a causal agent of downy mildew) on detached grapevine leaves, where dsRNA exhibited a curative effect by slowing disease progression in established infections[34].

Here, we used myrtle rust as a model system to build on our previous study, where we showed that dsRNA co-inoculated with urediniospores inhibited infection physiology of two rust fungi[26], to determine whether dsRNA could prevent future infections, as well as cure plants from severe existing infections. Rust fungi are obligate pathogens. While some taxa occur as latent infections, this has not been demonstrated for *A. psidii*, in which the presence of diseased vs healthy leaves is adequate to determine whether the fungus is present on the host[35]. In this study, the terms 'cure' or 'curative' denote the application of dsRNA after *A. psidii* infection resulting in significantly reduced disease symptoms and a restoration of plant health. 'Prevent' or 'preventative' signifies dsRNA application before infection leading to significant prevention or inhibition of myrtle rust symptoms.

Our overall aim was to improve plant health and recovery, following infection by *A. psidii*, with the eventual goal of application in conservation efforts. We tested hypotheses that *A. psidii*-specific dsRNA applied to leaves would (i) prevent infection, and (ii) cure established infections by inhibiting growth of rust fungi *in planta*. We used three time points in the disease cycle for curative assays, as well as a preventative time point, and assessed treatments by disease establishment, coverage, plant health, and indicators of photosynthesis to answer whether application of dsRNA can be translated in conservation efforts. Here we show that dsRNA, applied preventatively or curatively, inhibits myrtle rust disease and improves plant growth and recovery following severe infections.

## Results

**dsRNA targeting *A. psidii* can prevent disease development when applied prior to infection.** We tested whether pre-treatment of plants with *A. psidii*-specific dsRNA was protective against myrtle rust infection. When sprayed onto trees 48 h prior to infection, *translation elongation factor 1-a* (*EF1-a*), and *beta-tubulin* (*β-TUB*) dsRNA treatments provided significant protection against myrtle rust (Fig. 1). At 2 weeks post-infection, *EF1-a*-treated trees exhibited significantly lower disease coverage as compared to the negative and non-specific dsRNA (*green fluorescent protein* (*GFP*)) controls (*EF1-a*/-dsRNA: $p = 6.2791e-05$; *EF1-a*/*GFP*: $p = 0.0001$), as well as large (>0.8) effect sizes, as calculated by Hedge's g (*g*) (Supplementary Table 1, *EF1-a*/-dsRNA: $g = 3.81$; *EF1-a*/*GFP*: $g = 3.44$)[36,37]. Similarly, *β-TUB*-treated trees had significantly reduced disease coverage (*β-TUB*/-dsRNA: $p = 0.0001$; *β-TUB*/*GFP*: $p = 0.0002$) and large effect sizes (*β-TUB*/-dsRNA: $g = 3.38$; *β-TUB*/*GFP*: $g = 3.44$) compared to negative and non-specific controls (Fig. 1, Supplementary Figure 1, Supplementary Table 1).

We used scanning electron microscopy (SEM) to visualise infection on the leaf surface in preventative treatments. Urediniospores germinating on control leaves (sprayed with $H_2O$ or *GFP* dsRNA) had long germ tubes terminating in appressoria (as expected in 0–12 h, Fig. 2c), whereas spores on treatment leaves (sprayed with *EF1-a* or *β-TUB* dsRNA) had inhibited germination and stunted germ-tube development in germinated spores (Fig. 2c). These data support a hypothesis that dsRNA can protect host plants from infection by rust fungi on the leaf surface and confirm results previously observed in vitro[26].

**dsRNA targeting *A. psidii* can cure an established infection.** To test whether *A. psidii*-specific dsRNA could cure established infections, we applied dsRNA to infected plants at defined time points in the disease cycle. These time points were chosen based on the infection biology of *A. psidii* on *S. jambos*. At 24 h post-infection, urediniospores have germinated, appressoria have penetrated the leaf, infective hyphae are observed and haustoria have established but there are no macroscopic symptoms visible to the naked eye (Fig. 2a, b). At 6 days post-infection, the first

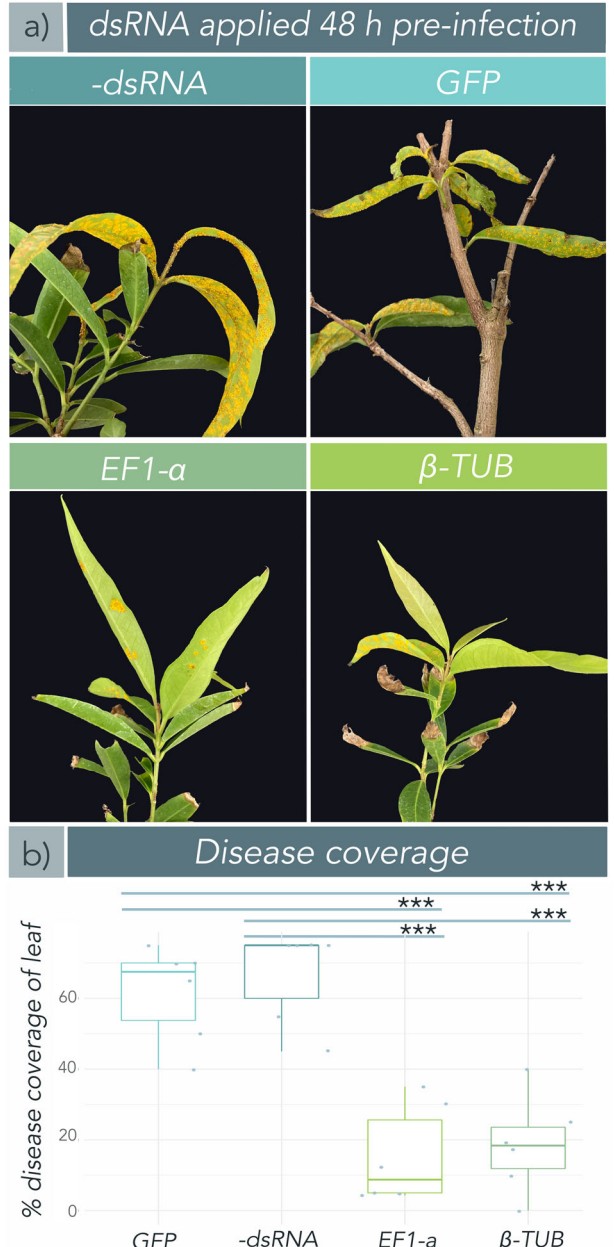

**Fig. 1 Exogenous application of *Austropuccinia psidii*-specific double-stranded RNA (dsRNA) provides significant protection against myrtle rust when applied 48 h prior to infection.** One-to-two-year-old *Syzygium jambos* trees ($n = 6$) grown under glasshouse conditions were treated with nuclease-free $H_2O$ (negative control), a non-specific dsRNA control (*green fluorescent protein* (*GFP*)), or *A. psidii*-specific dsRNAs *beta-tubulin* (*β-TUB*), or *transcription elongation factor* (*EF1-a*) at 100 ng/μL and challenged with *A. psidii* urediniospore inocula 48 h post-dsRNA treatment. Disease was assessed two weeks post-inoculation. **a** Photo comparison of *S. jambos* trees, one from each treatment and control group. -dsRNA and GFP control trees showed severe symptoms of myrtle rust, whereas EF1-a and β-TUB-treated plants showed significantly reduced symptoms. Photos were taken two weeks post-inoculation. **b** Box plot with superimposed scatter of diseased tissue as mean ($n = 6$) percent (%) coverage of leaf. Each biological replicate includes all young (susceptible) growth in the disease assessment (usually four to eight leaves). Disease was measured automatically using the Leaf Doctor application[57] two weeks post-inoculation. EF1-a and β-TUB dsRNAs significantly reduced diseased portions of leaves, as compared to -dsRNA and GFP controls. Significance is represented by asterisks (* = <0.05, ** = <0.01, *** = <0.001 (Welch's t-test)). Bars represent standard error of the mean. Figure was made in R 4.0.3[59].

*β-TUB/-dsRNA: g = 2.64, β-TUB/GFP: g = 2.90).* In addition to the reduction in disease coverage, plants treated with dsRNA 24 h post-inoculation also displayed a significant improvement in plant health at 6 weeks post-infection as compared to negative and non-specific controls (Fig. 3c, d; *EF1-a/-*dsRNA: *p = 0.00415*; *EF1-a/GFP: p = 0.00557*, *β-TUB/-dsRNA: p = 0.0096*; *β-TUB/GFP: p = 0.0145*; Table 1), and large effect sizes were again observed between groups (Supplementary Table 1; *EF1-a/-*dsRNA: *g = 2.21*; *EF1-a/GFP: g = 2.02, β-TUB/-dsRNA: g = 1.84, β-TUB/GFP: g = 1.64*). We further quantified plant photosynthetic performance through measurement of chlorophyll fluorescence parameters (Fv/Fm) and stomatal conductance ($g_{sw}$). An upward trend was observed from control to treatment groups in both Fv/Fm and $g_{sw}$, however, these trends were not significant for the dsRNA treatments administered at 24 h post-inoculation (Fig. 4).

*dsRNA treatment 6 days post-infection.* EF1-a and β-TUB dsRNA treatments were still effective when the application delay was extended to 6 days post-infection. We observed decreased disease coverage on *A. psidii* dsRNA-treated trees as compared to negative and non-specific controls when assessed at 14 days post inoculation (Fig. 3a, b, Supplementary Figure 1; *EF1-a/-*dsRNA: *p = 0.0016*; *EF1-a/GFP: p = 0.0023, β-TUB/-dsRNA: p = 0.0044; β-TUB/GFP: p = 0.0045*). This decrease in disease coverage on dsRNA-treated plants was further demonstrated by large effect sizes between control and treatment groups (Supplementary Table 1; *EF1-a/-*dsRNA: *g = 3.29*; *EF1-a/GFP: g = 2.48; β-TUB/-dsRNA: g = 4.12; β-TUB/GFP: g = 2.98*). Plants treated curatively with *β-TUB* or *EF1-a* dsRNA at first sign of pustule development at 6 days post-infection also significantly improved in measures of qualitative plant health observable at 6 weeks post-infection, as compared to the negative and non-specific controls (Fig. 3c, d; *EF1-a/-*dsRNA: *p = 0.0108*; *EF1-a/GFP: p = 0.0043, β-TUB/-dsRNA: p = 0.0005*; *β-TUB/GFP: p = 0.0017*). dsRNA treatments at 6 days post-infection resulted in a large effect size in plant health at 6 weeks post-infection in control versus treated plants (Supplementary Table 1; *EF1-a/-*dsRNA: *g = 1.78*; *EF1-a/GFP: g = 1.55; β-TUB/-dsRNA: g = 2.81; β-TUB/GFP: g = 2.27*). A significant increase in $g_{sw}$ was observed between both treatment and control groups (Fig. 4a; *EF1-a/-*dsRNA: *p = 0.0412*; *EF1-a/*

symptoms of *A. psidii* infection can be observed as flecking and small pustule development on *S. jambos*, and two-photon microscopy (TPEF) indicates that haustoria and intercellular hyphae are abundant (Fig. 2a, b, d). At 14 days post-infection, the disease cycle has completed. Plants are fully symptomatic and sori and urediniospores are present on the leaf surface (Fig. 2b, e). Thus, time points of 24 h, 6 days and 14 days after infection were chosen as key points at which to test the ability of dsRNA to cure an infection.

*dsRNA treatment 24 h post-infection.* EF1-a and β-TUB dsRNA significantly reduced disease coverage on *S. jambos* when applied 24 h post-inoculation, as compared to the negative and non-specific controls when assessed 14 days post infection (Fig. 3a, b, Supplementary Figure 1; *EF1-a/-*dsRNA: *p = 0.0042*; *EF1-a/GFP: p = 0.0007, β-TUB/-dsRNA: p = 0.0154, β-TUB/GFP: p = 0.00523*). Large effect sizes were observed between control and treatment groups, with respect to disease coverage (Supplementary Table 1; *EF1-a/-*dsRNA: *g = 2.38*; *EF1-a/GFP: g = 2.65,*

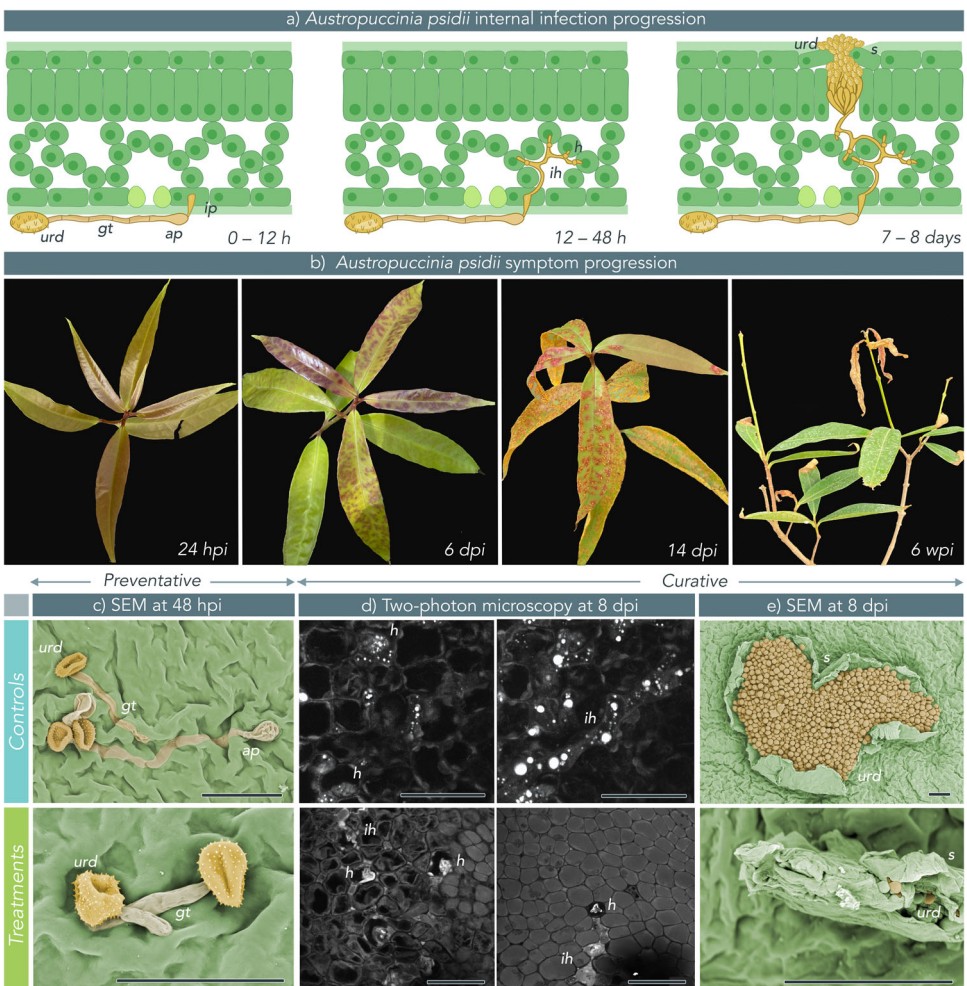

**Fig. 2 Infection cycle and disease progression of *Austropuccinia psidii* causing myrtle rust on double-stranded RNA (dsRNA) treated and untreated *Syzygium jambos* plants.** Leaves were treated with nuclease-free $H_2O$ (negative control), a non-specific dsRNA (*green fluorescent protein* (*GFP*)), or *beta-tubulin* (*β-TUB*), or *transcription elongation factor* (*EF1-a*) *A. psidii*-specific dsRNAs at 100 ng/μL. Plants were either challenged with *A. psidii* urediniospore inocula 48 h post-dsRNA treatment and sampled 48 h post-infection (hpi), or dsRNA was applied 6 days post-infection (dpi) and plants were sampled 48 h post-dsRNA treatment. **a** Internal infection progression of *A. psidii* with timing based on myrtle rust symptom development in *S. jambos*. **b** Disease progression of *A. psidii* on *S. jambos* over 6 weeks, showing key points in the infection cycle: 24 hpi = germination and penetration of leaf surface; 6 dpi = first symptoms *in planta* (red flecking from anthocyanin production in response to infection); 14 dpi = established infection and completion of a single disease cycle through production of new urediniospores; 6 weeks post-infection (wpi) = die back and defoliation as a result of severe infection. **c** Scanning electron microscopy (SEM) micrographs of germinating *A. psidii* urediniospores on preventative control (*GFP*) and treated (*β-TUB*) *S. jambos* plants. Scale bar = 50 μm. **d** Two-photon confocal micrographs of *A. psidii* intercellular hyphae and haustoria in curative control (*GFP*) (Top left and right) and treated (*EF1-a*) (bottom left and right) *S. jambos* leaves. For both samples, the left-hand image is taken at a depth approximately half-way through the leaf and the right-hand image is taken closer to the adaxial leaf surface. Scale bar = 50 μm. **e** SEM micrographs of developing sori in control (-dsRNA) and treated (*EF1-a*) *S. jambos* plants. Scale bar = 40 μm. Urediniospores (urd), germ tube (gt), appressorium (ap), infection peg (ip), haustorium (h), invasive hyphae (ih).

*GFP*: $p = 0.0148$, *β-TUB*/-dsRNA: $p = 0.0238$; *β-TUB*/*GFP*: $p = 0.0049$). There was an upward trend in Fv/Fm from control to treatment groups, however, the changes were not significant for the dsRNA treatments administered at 6 days post-inoculation (Fig. 4b).

We sampled leaves 8 days post-infection and 48 h post-dsRNA treatment and used TPEF to examine whether rust fungi had established in plant tissue. Indeed, haustoria and intercellular hyphae were seen in both control and treatment groups (Fig. 2d). The same samples were imaged with SEM to observe the presence of new disease symptoms on the leaf surface. On control leaves (-dsRNA or *GFP* dsRNA-treated) abundant sori, with several hundred new urediniospores in each sorus, were observed on all samples (Fig. 2e). Samples taken from treatment leaves (*EF1-a* and *β-TUB*) showed few or no sori. When sori were observed,

they were malformed and did not produce urediniospores (Fig. 2e).

*dsRNA treatment 14 days post-infection.* B-TUB and *EF1-a* dsRNA-treated plants showed significantly improved health and growth when measured 6 weeks post infection compared to negative and non-specific controls when dsRNA treatments were applied as late as 14 days post-infection, when myrtle rust infections were completely established (Fig. 3c, d, Supplementary Figure 1; *EF1-a*/-dsRNA: $p = 0.0021$; *EF1-a*/*GFP*: $p = 0.0043$, *β-TUB*/-dsRNA: $p = 0.0183$; *β-TUB*/*GFP*: $p = 0.0306$). Large effect sizes between control and treatment groups further demonstrated increases in plant health, when treatment occured at 14 days post-infection (Supplementary Table 1; *EF1-a*/-dsRNA: $g = 2.54$; *EF1-a*/*GFP*: $g = 2.46$; *β-TUB*/-dsRNA: $g = 1.68$; *β-TUB*/*GFP*: $g = 1.55$).

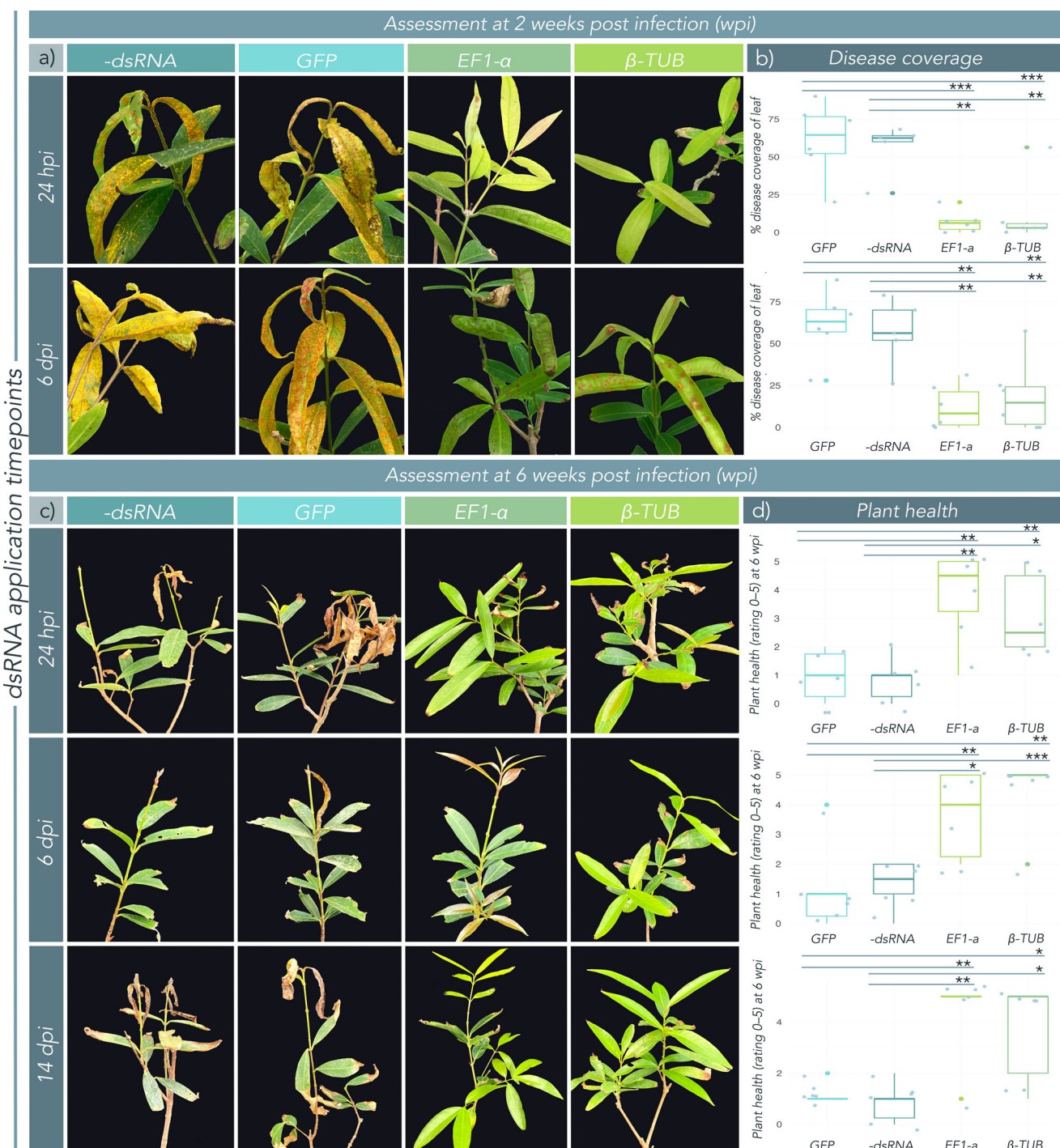

**Fig. 3 Exogenous application of *Austropuccinia psidii*-specific double-stranded RNA (dsRNA) cures myrtle rust infection at 24 h and 6 days post-infection.** *Syzygium jambos* trees ($n = 6$) were infected with *A. psidii* urediniospore inocula and treated with a nuclease-free $H_2O$ (negative control), a non-specific dsRNA control (*GFP* dsRNA), or *beta-tubulin (β-TUB)*, or *transcription elongation factor (EF1-a) A. psidii*-specific dsRNAs at a concentration of 100 ng/μL at 24 h, 6 days, and 14 days post-infection. **a** Photo comparison shows *S. jambos* plants inoculated with *A. psidii* and treated with *A. psidii*-specific dsRNA 24 h post-infection (hpi) or 6 days post-infection (dpi). Plants were photographed two weeks post-inoculation (wpi). **b** Boxplots with superimposed scatter of diseased tissue as mean ($n = 6$) percent (%) coverage of leaf. Each biological replicate includes all young (susceptible) growth in the disease assessment (usually four to eight leaves). Disease was measured automatically using the Leaf Doctor application[57] at 2 wpi. *EF1-a* and *β-TUB* dsRNAs significantly reduced diseased portions of leaves, as compared to –dsRNA and *GFP* controls. **c** Photo comparison shows *S. jambos* plants inoculated with *A. psidii* and treated with *A. psidii*-specific dsRNA 24 h, 6 days, or 14 days post-infection. Plants were photographed 6 wpi. **d** Boxplots with superimposed scatter of mean ($n = 6$) qualitative plant health. Plants were assessed according to the qualitative plant health scale (Table 1), developed in this study. *EF1-a* and *β-TUB* dsRNAs significantly increased qualitative plant health, as compared to -dsRNA and *GFP* controls. Significance in all boxplots is represented by asterisks (* = <0.05, ** = <0.01, *** = <0.001 (Welch's t-test)). Bars represent standard error of the mean. Figures (**b**) and (**d**) were made in R 4.0.3[59].

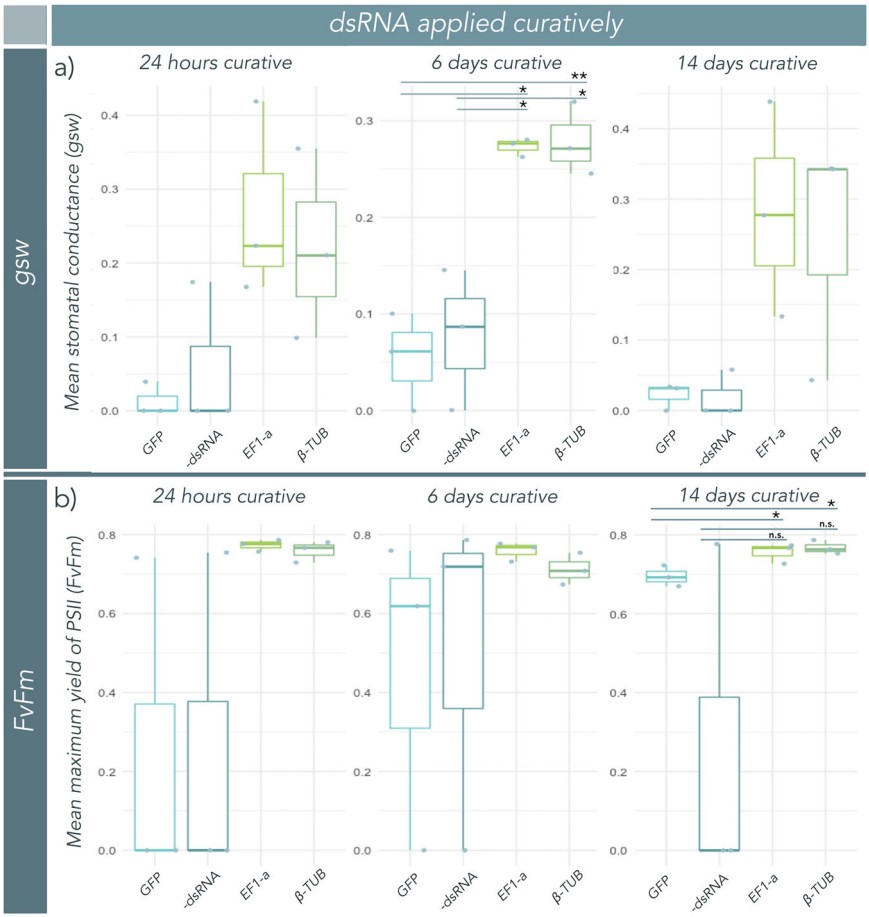

**Fig. 4 Measurements of stomatal conductance ($g_{sw}$) and maximum quantum yield of photosystem II (Fv/Fm) indicate improved plant photosynthetic performance of curatively treated *Syzygium jambos* plants.** $G_{sw}$ and Fv/Fm were measured 6 weeks post-infection for all control (-dsRNA or *GFP*) and treated (*EF1-a* and *β-TUB*) plants, which received dsRNA at multiple time points (24 h, 6 days, and 14 days) post-infection. **a** Boxplots with superimposed scatter of mean (*n* = 3) $g_{sw}$ for each treatment group at each time point. Treatment with *EF1-a* or *β-TUB* dsRNA 6 days post-infection resulted in a significant increase in mean $g_{sw}$ 6 weeks post-infection, as compared to both control groups. **b** Boxplots with superimposed scatter of mean (*n* = 3) Fv/Fm for each treatment group at each time point. Treatment with *EF1-a* or *β-TUB* dsRNA 14 days post-infection resulted in a significant increase in Fv/Fm at 6 weeks post-infection as compared to *GFP* controls, but not compared to -dsRNA controls. Significance in all boxplots is represented by asterisks (* = <0.05, ** = <0.01, *** = <0.001 (Welch's t-test)). Bars represent standard error of the mean. Graphs were made in R 4.0.3[59].

The upward trend in $g_{sw}$ was not significant at this time point for either dsRNA treatment groups, however, both treatment groups showed a significant increase in Fv/Fm as compared to the *GFP* control at 6 weeks post-dsRNA treatment (Fig. 4; *EF1-a/GFP*: $p = 0.0455$, *β-TUB/GFP*: $p = 0.02194$).

## Discussion
Preventative treatments are useful in annual crop pathosystems with coevolved boom-bust cycles to protect crops from short-term diseases, ensuring maximum yield outputs each year[38,39]. In contrast, perennial trees have long generation times, and pathogens can infect at any stage and must be managed at different stages of the disease cycle. The long-lived nature of perennials also carries an increased risk of new-encounter pathogens with short life cycles or the ability to change rapidly[13]. New treatment methods are needed that can tackle these concerns, especially in natural environments and forestry settings. We previously showed that *A. psidii*-specific dsRNA could prevent myrtle rust disease when co-applied to leaves with the urediniospore inoculum[26]. Here, we applied dsRNA to leaves pre- and post-infection and show that exogenously applied dsRNA can act both preventatively and curatively against myrtle rust disease, with

dsRNA-treated plants fully recovering from severe infection. We assessed disease and plant health through disease coverage measurements, a newly developed qualitative plant health scale (Table 1), effect sizes (Hedge's g), quantitative measurements of photosynthetic capacity, as well as TPEF and SEM, and demonstrate that dsRNA treatment improves the ongoing health of perennial plants.

Our work, together with other studies, have shown that fungi can take up dsRNA from the environment[26,40,41]. Further, dsRNAs may also be taken up by the fungus internally within a plant[42]. In fungi, dsRNA is hypothesised to be taken up through clathrin-mediated endocytosis[43], however, studies in this field are limited and there is still a significant knowledge gap in fungal dsRNA uptake[44,45]. Qiao et al. (2021) observed efficient uptake of dsRNA in several important plant pathogens, and in the case of *Phytophthora infestans*, uptake success varied between developmental stages of the same pathogen[40]. McRae et al. (2023) also demonstrated uptake of dsRNA by spores in the powdery mildew pathogen, *Golovinomyces orontii* MGH1[41]. In the present study, we build on previous observations of in vitro dsRNA uptake to show that dsRNA is having a similar inhibitory effect on urediniopsore germination and infection on leaves. We also show that haustoria and intercellular hyphae are likely able to take up

**Table 1 A qualitative scale used to rate overall plant health based on the number of new leaves per node.**

| Rating | Characteristics |
| --- | --- |
| 0 | Dead |
| 1 | Alive, but with no new growth |
| 2 | At least 1 new leaf, size of new leaves is ≥50% of the average size of existing leaves |
| 3 | 1–2 new leaves per node, size of new leaves is ≥50% of the average size of existing leaves |
| 4 | At least 2 new leaves per node, size of new leaves is ≥75% of the average size of existing leaves |
| 5 | At least 3 new leaves per node, size of new leaves is ≥75% of the average size of existing leaves |

dsRNA, as these structures could be observed 8 dpi and would have therefore been present at the 6 dpi curative time point (Fig. 2). This supports a hypothesis that dsRNA acts on urediniospores, haustoria and/or intercellular hyphae (Fig. 3) and suggests that dsRNA can be used as a treatment option in suited pathosystems even after disease is well established.

Our plant health assessments and LI-COR measurements on control plants demonstrate the cumulative impact of disease on plants that decrease their investment in photosynthesis from uninfected leaves. Rust fungi infection impacts plant health in part by limiting photosynthetic activity, whether through decreased available area on infected leaf surfaces, defoliation, or other physiological responses[46]. The myrtle rust pathosystem causes attrition in continued infections over time, and our measures of photosynthesis show that plants withstanding long-term infections have reduced growth and lose overall photosynthetic capacity (Figs. 3 and 4). In severe myrtle rust infections, plants lose highly infected and necrotic young leaves and their photosynthetic capacity relies on mature leaves that have survived infection. We developed a plant health rating system to score plant health and growth after plants are given time to recover (Table 1) and quantified that plants treated with *A. psidii*-specific dsRNA at all curative time points have increased growth 6 weeks after infection compared to their untreated counterparts (Fig. 3). The enhanced recovery and production of new growth in treated plants allows for increased photosynthetic capacity and improvements in overall health.

As further quantification, our measures of Fv/Fm and $g_{sw}$ as indicators of photosynthetic activity in control and treated plants show that curative application of dsRNA may improve plant photosynthetic performance, impacting different measures significantly at different time points (Fig. 4). A previous study of biotic impacts on photosynthesis found a significant correlation between $g_{sw}$ and net photosynthetic rate (Pn); Zhang et al. (2022) hypothesised that a decline in Fv/Fm and $g_{sw}$ causes a decrease in Pn, in response to pathogen infection. Since stomata allow photosynthesis-enabling gas exchange, their closure decreases $g_{sw}$ and can inhibit photosynthesis[47]. Similarly, Fv/Fm measurements are often used for early detection of stress in plants, and a lower value may be indicative of photoinhibition[48,49]. Here, we see a consistent trend of higher $g_{sw}$ and Fv/Fm values in dsRNA treated compared to untreated plants. In some cases, these measures were statistically significant (Fig. 4). While we observed contrasting mean Fv/Fm scores at 14 dpi in the two control groups (Fig. 4b), this is likely to have occurred due to the lower and more variable plant health ratings for -dsRNA compared to *GFP* plants at 14 dpi (Fig. 3d), which is reflected in a lower Fv/Fm score for this control group. These results align with other studies of biotic impacts on rust fungi[46,50] and further support our curative results, suggesting that plants treated curatively with dsRNA may have a higher photosynthetic capacity.

Curative treatments may provide further relief by preventing or lowering production of viable new inoculum and inhibiting epidemic disease cycles. In our study, plants sprayed at 6 dpi were harvested at 8 dpi and imaged with SEM to visualise the impacts of curative dsRNA on emerging sori (Fig. 2). Control plants showed abundant urediniospore-filled sori on the leaf surface at 8 dpi, whereas few or no sori were seen on treated plants at the same stage in the infection cycle. In cases where a sorus could be seen on a treated leaf, it appeared malformed and collapsed, with only a small number of urediniospores visible (Fig. 2). This result suggests that when dsRNA is applied at 6 dpi, the point of earliest symptom development in the form of red flecking on the leaf, it may be taken up into early-forming sori or intercellular hyphae, interfering with regular formation of the sorus. However, further quantification of the impacts of dsRNA on emerging sori will be needed to support this hypothesis, including sampling and viability tests of urediniospores produced in curatively treated plants.

Our study indicates that dsRNA acts preventatively to significantly reduce disease coverage of *A. psidii* when applied 48 h pre-infection (Fig. 3), which is consistent with the demonstrated persistence of naked dsRNA on leaf surfaces for 5–7 days[51]. There was a slight reduction in protection compared to our previous study where the dsRNA was co-applied with the urediniospores. Although dsRNA can persist on leaves for days, it does begin to degrade during this timeframe[26,51]. Therefore, it is likely that the lower protective effect was due to gradual dsRNA degradation on the leaf surface, resulting in a lower amount being available to the fungus at the time of inoculation. Prior assays in this pathosystem have also shown that effectiveness of dsRNA is proportional to dose. Results from this preventative assay, along with our previous findings of dsRNA uptake into *A. psidii* urediniospores[26], suggest that dsRNA is being taken up by urediniospores on the leaf surface and is not being taken up by *S. jambos* leaves[26]. However, given the curative results, it is possible that dsRNA can enter infected leaves through leaf openings formed by infection pegs[50,52], and be taken up by intercellular hyphae and/or haustoria.

Our curative and preventative findings provide a new solution to the more than 10-year epidemic of myrtle rust in Australia, which has catastrophically brought at least three species of Myrtaceae to the brink of extinction[5], and predicts the loss of at least 16 more species in a single generation[14]. We have shown that plants at any stage of infection can recover from disease when treated with dsRNA targeting *A. psidii*. This treatment option does not need a carrier to protect dsRNA when applied to plants that are already infected, and dsRNAs are designed to specifically target *A. psidii* barcoding genes[26], so as to prevent any off-target effects. Additionally, large-scale microbial-based dsRNA production pipelines now allow for cost-effective synthesis of dsRNA, down from $12,500 USD per gram in 2008, to as little as $1 USD per gram in 2023[53–55]. Thus the only limiting factor is the practicality of applying dsRNA to large trees or in natural ecosystems. Despite this limitation, this approach still has the potential for immediate applicability in the nursery industry and ex situ conservation, and to protect or cure culturally significant trees infected by *A. psidii*. Further delays in deployment of treatments such as these may result in an increased need for the use of harmful fungicides in industry, or the loss of culturally significant trees, and environmentally significant species.

Beyond its single-pathosystem applications, curative dsRNA is a tool in the arsenal of long-term disease management, especially in forestry industries, and in the conservation of perennial trees, where the aim is to encourage the ongoing health and growth of a tree over long periods, rather than to improve short-term yields over a single growing season. Preventative RNAi combined with early disease detection, surveillance using artificial intelligence (AI), and remote sensing[56] also has significant potential in agriculture to manage pathogens and reduce crop losses.

## Methods

**dsRNA synthesis, inoculum preparation, and infection conditions**. dsRNAs for all treatments and non-specific controls were synthesised according to Degnan et al.[26] dsRNA was synthesised in vitro using the HiScribe T7 High Yield RNA Synthesis Kit (New England BioLabs) with T7 PCR products as templates. Template sequences were amplified from *A. psidii* cDNA using Phusion polymerase and primers containing T7 promoter sequences. Once synthesised, dsRNA was purified with TRIzol (Thermofisher Scientific).

Urediniospores of *A. psidii* were wild harvested from trees in South-East Queensland by shaking sporulating leaf infections into a paper bag. Urediniospores were dried in a desiccator for 48 h and aliquoted into 200 μl PCR tubes for storage at −80 °C. To prepare inocula, urediniospores were suspended in sterile distilled water with 0.05% Tween 20 at a concentration of $10^6$ spores/ml. Urediniospores were allowed to rehydrate in inocula for 30 minutes prior to use. For all assays, urediniospores were germinated in planta in a controlled environment room (CER) at 18 °C, 75% relative humidity, and with no lights for 24 h.

**LI-COR measurements and calibrations**. Three physiological response variables relating to plant photosynthetic performance were measured on treatment and control plants 6 weeks post-infection. Dark-adapted optimal PSII yield (Fv/Fm) and stomatal conductance ($g_{sw}$) were measured between 11:00 am and 2:00 pm on sunny days using a LI-600 Porometer/Fluorometer (LI-COR Biosciences, Inc., Lincoln, NE, USA). The youngest, fully-expanded leaves on each plant were selected. Fv/Fm was measured following a 30-min dark-adaption period. Measurements were taken at a constant flow rate of 150 μmol $s^{-1}$, with a flash intensity of 7000 μmol $m^{-2} s^{-1}$ (6000 μmol $m^{-2} s^{-1}$ for dark-adapted plants).

**Preventative plant assays**. One to two-year-old *S. jambos* trees were grown in a glasshouse and were watered once daily by hand, without wetting leaves. Trees were maintained with Nitrosol fertilisation at two-week intervals in the lead up to infection, they were not fertilised while experiments were active. Trees were sprayed on adaxial and abaxial leaf surfaces with ~500 μl per node (two leaves) of either nuclease-free (NF) $H_2O$ (negative control), or 100 ng/μl *EF1-a*, *β-TUB*, *GFP* (non-specific control) dsRNA in NF $H_2O$. Six biological replicates (trees) were used per treatment and 6–8 technical replicates (leaves) were measured per tree, depending on the number of new leaves susceptible to infection. Technical replicates from each tree were aggregated into one data point (biological replicate) for statistical analysis and in figures. After 48 h, all plants were spray-inoculated until covered, before the point of run-off, on adaxial and abaxial leaf surfaces with *A. psidii* inocula, prepared as above. After 24 h, plants were removed from the CER and returned to the glasshouse. After two weeks, disease incidence was assessed using Leaf Doctor[57] based on percent coverage of diseased tissue[26] and LICOR measurements were taken as above. Defoliated leaves (leaves that had fallen off

the plant due to high levels of infection) were given an automatic disease percent coverage of 75% (Fig. 2b).

**Curative plant assays**. *Syzygium jambos* trees were spray-inoculated with *A. psidii* inoculum. Tree maintenance, inocula preparation, and germination conditions were as above. Trees were sprayed with ~1 ml (or until young flush was covered) of either NF $H_2O$ (negative control) or 100 ng/μl *EF1-a, β-TUB* or *GFP* (non-specific control) dsRNA at 24 h, 8 days, or 14 days post-infection. These time points were selected based on their alignment with stages of the infection process: Early infection, development of infection pegs and establishment of haustoria (24 h); symptomatic, first pustules develop on leaf (8 days); established infection (14 days). Separate plants were used for each treatment and time point. Six biological replicates (trees) were used per treatment, per time point. Infections were monitored, and disease coverage was assessed at two weeks post-inoculation as above for 24 h and 8-day curative plants. Qualitative plant health was measured at 6 weeks post-infection (Table 1). LI-COR measurements were made on all plants at 6 weeks post-infection, as above.

**Scanning electron microscopy (SEM)**. *Austropuccinia psidii*-infected *S. jambos* leaves were imaged with SEM at 2 days post-infection/four days post-dsRNA application for preventative experiments, and at 8 days post-infection/two days post-dsRNA application for curative experiments. Leaves were sampled using a sterile disposable 5 mm biopsy punch and were immediately fixed in 2.5% glutaraldehyde in phosphate-buffered saline (PBS). Samples were washed twice with PBS in a Pelco Biowave on a cycle of 1 min on, 1 min off, 1 min on. Leaf samples were then dehydrated through a graded ethanol series of 10%, 30%, 50%, 70%, 90%, and twice at 100%, with a Biowave cycle of 1 min on, 1 min off, and 1 min on at each ethanol grade. Samples were then stepped into hexamethyldisilazane (HMDS) through Biowave cycling (1 min on, 1 min off, 1 min on) first at a 1:1 ratio with the solvent (100% ethanol) and then twice in 100% HMDS for final-stage drying. The majority of the HMDS was removed and samples were left to dry completely in the fume hood for ~1 h. Once dried, samples were mounted on aluminium stubs with two-sided carbon tape and sputter coated with Platinum at 5 nm thickness using a Safematic CCU-010 sputter coater. Following coating, leaves were imaged with a Hitachi TM400 Plus benchtop SEM at an accelerating voltage of 5 kv and a working distance of 9–11 mm. SEM images were manually coloured in Adobe Photoshop version 23.5.5.

**Two-photon microscopy (TPEF)**. *Austropuccinia psidii*-infected *S. jambos* leaves were imaged with TPEF at 8 days post-infection/2 days post-dsRNA application for curative experiments. Leaves were prepared for TPEF based on Sørensen et al. with several modifications[58]. Leaves were sampled using a sterile disposable 5 mm biopsy punch and transferred to a fixing and clearing solution of 0.15% (w/v) trichloroacetic acid in ethanol:chloroform (3:1, v/v) for 24 h. Samples were washed twice and stored in 50% ethanol. Directly prior to imaging, samples were incubated in NaOH for 30 minutes at RT, rinsed in distilled water, and incubated in 0.1 M Tris-HCl buffer (pH 7.5) for 30 min at RT. Leaves were stained for 5 min in 2 μM Wheat germ agglutinin (WGA)—fluorescein isothiocyanate (FITC) at RT, and washed four times in distilled water prior to imaging.

Samples were imaged with a Zeiss LSM confocal microscope and WGA-FITC-stained fungal material was excited by a TPEF laser at 800 nm with a detection wavelength of 415 nm and emission at 430 nm. Images were deconvoluted using

Microvolution. Six iterations were run with lateral spacing of 87.85 nm/pixel, axial spacing of 269.06 nm/splice, an objective refraction index of 1.52, and a sample refractive index of 1.3.

**Statistics and reproducibility.** Six biological replicates (trees) were used per treatment, with 6 to 8 technical replicates (leaves) per biological replicate. Technical replicates were aggregated into one data point (biological replicate) for statistical analysis and for data visualisation. Welch's two sample, two-tailed $t$ tests were computed in R Studio 1.3.1093[59] at a 95 percent confidence interval using the dplyr package and pairwise t-test function[60]. $P$ values of <0.05 were considered significant and >0.05 were considered not significant. Hedge's g was used to quantify to magnitude of difference, or effect size, between control and/or treatment groups. Hedge's g was computed in R Studio 1.2.1093[59]. Data were plotted in R Studio 1.3.1093[59] using box (geom_box) and scatter (geom_scatter) plots with ggplot2 package[60].

**Reporting summary.** Further information on research design is available in the Nature Portfolio Reporting Summary linked to this article.

## Data availability

Raw disease coverage scores, plant health ratings, Fv/Fm and $g_{sw}$ data, and all TPEF and SEM micrographs that support the findings of this study are publicly available on Zenodo at DOI 10.5281/zenodo.8219701. Source data for Fig. 1 are found in file preventative_diseasecoverage_rawdata.csv. Source data for Fig. 3 are found in files curative_24hpi_diseasecoverage_rawdata.csv, curative_24hpi_planthealth_rawdata.csv, curative_6dpi_diseasecoverage_rawdata.csv, curative_6dpi_planthealth_rawdata.csv, and curative_14dpi_planthealth_rawdata.csv. Source data for Fig. 3 are found in file licor_curative.csv.

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

## Acknowledgements

We thank Dr. Kathryn Green from the Centre for Microscopy and Microanalysis at the University of Queensland for her time, assistance and expertise in Scanning Electron Microscopy (SEM), and Dr. Deborah Barkauskas from the Institute for Molecular Bioscience Microscopy facility at the University of Queensland for conducting the two-photon confocal microscopy (TPEF), and deconvolution of images. R.M.D. thanks the Plant Biosecurity Research Initiative (PBRI) and program director Dr. Jo Luck for their support. We thank the Australian Plant Biosecurity Foundation (APBSF) for their support. A.S. was supported by an Advance Queensland Industry Research Fellowship. This research was partially supported by the Australian Research Council Research Hub for Sustainable Crop Protection (IH190100022) funded by the Australian Government.

## Author contributions

R.M.D. designed and conducted experiments, and R.M.D. and A.R.M. wrote the manuscript. A.S., A.R.M., L.S.S., J.R.S and D.M.G. designed experiments and edited the manuscript. L.S.S. developed, optimised, and provided experimental pathosystem set-up, and J.R.S. conducted LI-COR measurements and assisted in disease assessments. A.S. and A.R.M. provided principal supervision, and L.S.S., D.M.G., B.J.C. and N.M. assisted in supervision and edited the manuscript.

## Competing interests

The authors declare no competing interests.
