## [Peer Review File · Communications Biology]

Double-stranded RNA prevents and cures infection by rust fungiReviewers' comments:

Reviewer #1 (Remarks to the Author):

The manuscript reports their findings that dsRNAs targeting *Austropuccinia psidii* (causal agent of myrtle rust) Translation Elongation Factor 1 (EF1-alpha) and beta-Tubulin can provide both preventive and curative protection to *Syzygium jambos* trees. The manuscript is well written and data were carefully presented with proper statistical analysis. It is acceptable for publishing in the journal with some minor suggested revisions listed below to further strengthen manuscript:

1. The authors pointed out (lines 166-168) "Intercellular hyphae and haustoria were observed in both controls and treatments, some treatments showed reduced proliferation of intercellular hyphae, however these differences were difficult to quantify (Figure 2d)". This can be easily addressed by extracting DNA from your collected infected leaf tissues and measure the relative ratio of a fungal reference gene (such as actin or 18S) to a plant reference gene through real time PCR to assess the degree of fungal infection among different treatments. Adding these data can significantly strengthen this manuscript in my view.
2. A discussion or some kind of explanation on why the FvFm in the -dsRNA control in 14 day curative sample in Figure 4 behaved differently compared to GFP control would be necessary.
3. DsRNA application: in the method section (lines 323-324), you indicated "Trees were sprayed with ~1 ml (or until young flush was covered) of either nuclease free (NF) H2O....". This is not clear enough. Please indicate how much dsRNA were applied per tree or specify the volume since it could require several ml of dsRNA to achieve "until young flush was covered". Please also indicate whether both sides or only the top or underside of the leaves were sprayed with dsRNA. Do the same for the part of inoculation.
4. Stomatal conductance is written as $g(\text{subscript})\text{sw}$. Please be consistent throughout the manuscript. Please change "gsw" on lines 252, 255, 256, 257 and 260 to " $g(\text{subscript})\text{sw}$ ".
5. Line 264: "have a positive flow on effect preventing...". It seems something is missing or should be removed in this phrase. Please revise.

Reviewer #2 (Remarks to the Author):

This study evaluated the effects of dsRNA spray before or after *Austropuccinia psidii* infection on tree rust development. They found that dsRNA spray could prevent and cure rust fungi infection at different disease stages, including before and after rust fungi infection. Moreover, plant photosynthetic activities were improved after dsRNA treatments. It provides useful scientific information to control myrtle rust on infected trees. It is therefore recommended for publishing.

Minor comments

Figure 1: Enlarge the number of Y axes.

L159-162: Do they compare to a 24 curative?

L180: Compared to -dsRNA control according to Figure 4?

Figure 3: Does the n=6 present six tree plants or six leaves?

L282-284: evidence?

L305: Although dsRNA synthesis was described in previous paper, it is helpful to add a brief description here.

Curious if it was tested on the infected tree in the forest? What are the efficacy and the consistency?

Reviewer #3 (Remarks to the Author):

In the manuscript "Solutions for a plant disease epidemic: double-stranded RNA prevents and cures infection by rust fungi," Degnan et al. investigate a new means of preventing and treating myrtle rust disease in the plant *Syzygium jambos*. This is a worthy goal because perennial plant species in the family Myrtaceae are ecologically impacted by *Austropuccinia*, and their long lifespan and wild habitat make breeding for genetic resistance difficult. The proposed treatment uses exogenously applied double-stranded RNA to cause RNA interference, which disrupts expression of

fungal genes that are presumably essential for growth/development/sporulation. The molecular mechanism was not tested in this manuscript, but work published earlier this year by the same research group suggests that this may be happening, and several previous studies have demonstrated it in other pathosystems. In this study, plants that were pre-treated with dsRNA developed less severe infection when inoculated with the pathogen. Also, infected plants that were treated at 24h, 6 days, and 14 days after infection were healthier on average six weeks later. Photography, microscopy, and physiological measurements were used to support the treatment claims.

Overall, this manuscript has potential to make a good contribution to the literature. The findings are novel in that dsRNA has not been used extensively to treat fungal disease after visible symptoms begin. The chosen dsRNAs appear efficacious; the chosen metrics are relevant for plant disease; the manuscript is written clearly; and the images and micrographs are clearly presented. However, I have some concerns. My main issue is that treatment effect sizes are not sufficiently documented. For example, "prevention" as in Fig. 1 is a claim of clinically meaningful significance; statistical significance is only one part of that. Please estimate the effect size in the form of diseased area reduction, fewer lesions, smaller lesions, or some other relevant metric with units, and present it in the text. I also take issue with the frequent use of "cure" and "curative." My (perhaps uninformed) understanding of "cure" is more or less complete cessation of disease symptoms, ideally accompanied by a negative test for the pathogen itself. This is the common usage of the term (e.g. "penicillin cures some types of bacterial infection"). At present it seems that the words "treatment" or "inhibition" are more warranted. In Fig. 3 some plants may indeed be cured based on this stringent definition, although the effect is noisy and given the small sample size, 2-3 out of 6 plants look similar to controls. Here too it would strengthen the authors' claims to discuss effect sizes. Finally, some claims in the introduction do not seem to match what is shown in the results.

I support the publication of this manuscript provided that the authors revise and clarify some points. Line-by-line comments, questions, and suggestions are as follows.

48-49 Does forest management practice currently include/allow foliar fungicide treatment? Introduction and Discussion both mention that dsRNA could be applied in a wild setting, so it would be beneficial to have sentence describing current in situ conservation efforts.

68-70. "cure plants from severe fungal infections and prevent further infections" Please clarify. "Further" implies that the plants were first cured of infection, and then became more resistant to subsequent pathogen challenge. This study did not test that, and it is a different goal than than the aims to "(i) prevent infection, and ii) cure established infections" (line 70). Suggestion: introduce the aims in the order they are dealt with in the results. ^[SEP]

69 "Our overall aim was to improve plant performance and conservation outcomes" Doesn't seem accurate; conservation outcomes were not tested in this study. Suggestion: "Our overall aim was to improve measures of plant health with the eventual goal of application in conservation efforts."

86 Clear images with an obvious treatment effect. It would be nice to see a flat leaf image representing each treatment group. I assume the "Leaf Doctor" app takes flat images as the input, correct? This could be a supplemental.

88 *One-to-two-year old ^[SEP]

98 *asterisks

103 Did you try combining the two dsRNAs (EF1-a and b-TUB) 1:1 as a combination therapy? Not required but I'm curious if there might be synergy.

123 This sentence is also in Methods. It's fine if it appears only in Methods.

125-126 "dsRNA targeting *A. psidii* can cure an established infection" Suggestion: "dsRNA targeting *A. psidii* can effectively treat an established infection." The treatment strongly inhibits disease progression, but in my opinion this does not warrant the word cure without more explanation.

198 *asterisks

201-202 Standardize abbreviation for stomatal conductance. Sometimes it is lowercase with subscript and other times not.

209 *asterisks

264 "positive flow-on effect" Word choice — not sure what is meant here. Suggestions: knock-on effect / positive feedback / virtuous circle. Maybe there is a standard epidemiology term for this

kind of effect?

291 If you bring up the practicality of dsRNA application, it would help to compare it to current practices, if any. $100 \text{ ng RNA} / \mu\text{L} = 100 \text{ mg} / \text{L}$, correct? Can this be produced at a scale sufficient for use in the field or greenhouse? How does this compare to the cost of a conventional foliar fungicide? These questions are optional and beyond the scope of the current work, but could help to get people inspired if you advocate for more widespread dsRNA use.

295 "harmful fungicides in industry" If you bring up toxicity of conventional fungicides, you also need to mention potential off-target effects of dsRNA on benign and beneficial species. The target genes appear very conserved, and the text does not mention any effort to make the dsRNA sequences specific to *A. psidii*.

305 Please provide a citation number. I assume this is Ref. 25.

328-330 Please provide more detail on area measurements. Fig. 1 legend states that this is "mean ($n = 6$) percent coverage" of new growth leaves, and that there are 6-8 leaves per tree. Methods should specify how these measurements were aggregated. Were individual leaf measurements pooled for the leaves on a particular tree, which then became an individual dot on the box plot?

Response to Reviewers

Reviewer #1 (Remarks to the Author):

*The manuscript reports their findings that dsRNAs targeting *Austropuccinia psidii* (causal agent of myrtle rust) Translation Elongation Factor 1 (EF1-alpha) and beta-Tubulin can provide both preventive and curative protection to *Syzygium jambos* trees. The manuscript is well written and data were carefully presented with proper statistical analysis. **It is acceptable for publishing in the journal with some minor suggested revisions listed below to further strengthen manuscript:***

- 1. The authors pointed out (lines 166-168) “Intercellular hyphae and haustoria were observed in both controls and treatments, some treatments showed reduced proliferation of intercellular hyphae, however these differences were difficult to quantify (Figure 2d)”. This can be easily addressed by extracting DNA from your collected infected leaf tissues and measure the relative ratio of a fungal reference gene (such as actin or 18S) to a plant reference gene through real time PCR to assess the degree of fungal infection among different treatments. Adding these data can significantly strengthen this manuscript in my view.*

While we appreciate the reviewer’s suggestion, the purpose of our two-photon microscopy (TPEF) photographs in Figure 2 primarily serves to illustrate disease progression and the presence of infection structures, such as haustoria, at 8 dpi, showing that dsRNA is able to effectively treat myrtle rust symptoms even after the establishment of such structures. This is outlined in lines 203–205.

Although we suggested that proliferation of intercellular hyphae may be reduced in treatment groups, this was not the purpose of TPEF investigations. We have now re-worded this to clarify the purpose of the experiment and results. This now reads as “We sampled leaves 8 days post-infection and 48 hours post-dsRNA treatment and used TPEF to examine whether rust fungi had established in plant tissue. Indeed, haustoria and intercellular hyphae were seen in both control and treatment groups (Figure 2d)” (lines 204–205).

In summary, while we acknowledge the benefits of real-time PCR quantification in many experiments, we emphasize that our TPEF data serves to illustrate disease progression over time, presence of infection structures at this timepoint, and to confirm that later-stage curative dsRNA treatments are indeed effective even after the establishment of such infection physiology.

- 2. A discussion or some kind of explanation on why the Fv/Fm in the –dsRNA control in 14 day curative sample in Figure 4 behaved differently compared to GFP control would be necessary.**

We have added a sentence discussing how the Fv/Fm in -dsRNA and *GFP* controls corresponds to the variability (or lack of) in plant health in -dsRNA and *GFP* plants. As in, plant health data of 14 dpi -dsRNA plants are more variable than that of 14 dpi *GFP* plants. These differences in variability are also reflected in Fv/Fm of the same plants, at the same timepoint.

This reads as “While we observed contrasting mean Fv/Fm scores at 14 dpi in the two control groups (Figure 4b), this is likely to have occurred due to the lower and more variable plant health ratings for -dsRNA compared to *GFP* plants at 14 dpi (Figure 3d), which is reflected in a lower Fv/Fm score for this control group” (lines 316-319).

- 3. DsRNA application: in the method section (lines 323-324), you indicated “Trees were sprayed with ~1 ml (or until young flush was covered) of either nuclease free (NF) H₂O.....”. This is not clear enough. Please indicate how much dsRNA were applied per tree or specify the volume since it could require several ml of dsRNA to achieve “until young flush was covered”. Please also indicate whether both sides or only the top or underside of the leaves were sprayed with dsRNA. Do the same for the part of inoculation.**

We have added clarification on dsRNA volumes, and dsRNA and *A. psidii* inocula spray methods in these sections. This now reads as “Trees were sprayed on adaxial and abaxial leaf surfaces with ~500 µl per node (two leaves) of either nuclease free (NF) H₂O (negative control), or 100 ng/µl *EF1-a*, *β-TUB*, *GFP* (non-specific control) dsRNA in NF H₂O” (lines 400–402) and “Six biological replicates (trees) were used per treatment and 6 – 8 technical replicates (leaves) were measured per tree, depending on the number of new leaves susceptible to infection. Technical replicates from each tree were aggregated into to one data point (biological replicate) for statistical analysis and in figures. After 48 hours, all plants were spray-inoculated until covered, before the point of run-off, on adaxial and abaxial leaf surfaces with *A. psidii* inocula, prepared as above” (lines 402–407).

- 4. Stomatal conductance is written as $g(\text{subscript})sw$. Please be consistent throughout the manuscript. Please change “gsw” on lines 252, 255, 256, 257 and 260 to “ $g(\text{subscript})sw$ ”.**

We have now corrected gsw to $g(\text{subscript})sw$ on these lines.

- 5. Line 264: “have a positive flow on effect preventing...”. It seems something is missing or should be removed in this phrase. Please revise.**

This has been reworded to: “Curative treatments may provide further relief by preventing or lowering production of viable new inoculum and inhibiting epidemic disease cycles.” (Line 324-325)

Reviewer #2 (Remarks to the Author):

This study evaluated the effects of dsRNA spray before or after Austropuccinia psidii infection on tree rust development. They found that dsRNA spray could prevent and cure rust fungi infection at different disease stages, including before and after rust fungi infection. Moreover, plant photosynthetic activities were improved after dsRNA treatments. It provides useful scientific information to control myrtle rust on infected trees. It is therefore recommended for publishing.

Minor comments

1. Figure 1: Enlarge the number of Y axes.

The Y axis text in Figure 1 has been enlarged to improve readability.

Figure 1. Exogenous application of *Austropuccinia psidii*-specific double-stranded RNA (dsRNA) provides significant protection against myrtle rust when applied 48 hours prior to infection.

2. L159-162: Do they compare to a 24 curative?

There is no significant difference in g(subscript)sw of 6-day curative dsRNA treatments (*EF1-a* and *B-TUB*) when compared to 24-hour curative dsRNA treatments (*EF1-a* and *B-TUB*). We mention in the text that while there is an upward trend from control to treatment groups at all timepoints, the difference was only significant at the 6-day curative timepoints. As the trends are similar across all three timepoints, we anticipate that the changes were not significant in 24-hour and 14-day curative treatments due to greater variation in the data.

3. L180: Compared to -dsRNA control according to Figure 4?

The difference is significant in comparison to the *GFP* control, but not compared to the -dsRNA control. We have now edited this figure, by removing the lines and moving the significance asterisks to sit over the treatment with significant difference to the *GFP* non-specific control (Figure 4). We have also added a sentence in the figure caption – consistent with our approach in other figure captions – to explain the significance. This now reads as “Treatment with *EF1-a* or β -*TUB* dsRNA 14 days post-infection resulted in a significant increase in Fv/Fm at 6 weeks post-infection as compared to *GFP* controls, but not compared to -dsRNA controls” (lines 259–261).

Figure 4. Measurements of stomatal conductance (g_{sw}) and maximum quantum yield of photosystem II (Fv/Fm) indicate improved plant photosynthetic performance of curatively treated *Syzygium jambos* plants. g_{sw} and Fv/Fm were measured 6 weeks post-infection for all control (-dsRNA or *GFP*) and treated (*EF1-a* and β -*TUB*) plants, which received dsRNA at multiple timepoints (24 hours, 6 days, and 14 days) post-infection. (a) Boxplots with superimposed scatter of mean ($n=3$) g_{sw} for each treatment group at each timepoint. Treatment with *EF1-a* or β -*TUB* dsRNA 6 days post-infection resulted in a significant increase in mean g_{sw} 6 weeks post-infection, as compared to both control groups. (b) Boxplots with superimposed scatter of mean ($n=3$) Fv/Fm for each treatment group at each timepoint. Treatment with *EF1-a* or β -*TUB* dsRNA 14 days post-infection resulted in a significant increase in Fv/Fm at 6 weeks post-infection

as compared to *GFP* controls, but not compared to –dsRNA controls. Significance in all boxplots is represented by asterisks (* = <0.05, ** = <0.01, *** = <0.001 (Welch's t-test)). Bars represent standard error of the mean. Graphs were made in R 4.0.3⁹.

4. *Figure 3: Does the n=6 present six tree plants or six leaves?*

N=6 represents six trees. In the instance of disease coverage assessments, six to eight leaves were assessed per tree as technical replicates. This has now been clarified in the methods on lines 402–404. This reads as “Six biological replicates (trees) were used per treatment and 6 – 8 technical replicates (leaves) were measured per tree, depending on the number of new leaves susceptible to infection”.

5. *L282-284: evidence?*

We have added references to support our hypothesis dsRNA stability, degradation, and availability on the leaf surface to line 339.

6. *L305: Although dsRNA synthesis was described in previous paper, it is helpful to add a brief description here.*

We have added a brief description of dsRNA synthesis on lines 378–381. This now reads as “dsRNA was synthesised *in vitro* using the HiScribe T7 High Yield RNA Synthesis Kit (New England BioLabs) with T7 PCR products as templates. Template sequences were amplified from *A. psidii* cDNA using Phusion polymerase and primers containing T7 promoter sequences. Once synthesised, dsRNA was purified with TRIzol (Thermofisher Scientific)”.

7. *Curious if it was tested on the infected tree in the forest? What are the efficacy and the consistency?*

dsRNA treatments have not yet been tested against *A. psidii* in natural forest settings, although we agree that this is certainly of interest and is a focus for future work.

Reviewer #3 (Remarks to the Author):

In the manuscript “Solutions for a plant disease epidemic: double-stranded RNA prevents and cures infection by rust fungi,” Degnan et al. investigate a new means of preventing and treating myrtle rust disease in the plant *Syzygium jambos*. This is a worthy goal because perennial plant species in the family Myrtaceae are ecologically impacted by *Austropuccinia*, and their long lifespan and wild habitat make breeding for genetic resistance difficult.

The proposed treatment uses exogenously applied double-stranded RNA to cause RNA interference, which disrupts expression of fungal genes that are presumably essential for

growth/development/sporulation. The molecular mechanism was not tested in this manuscript, but work published earlier this year by the same research group suggests that this may be happening, and several previous studies have demonstrated it in other pathosystems. In this study, plants that were pre-treated with dsRNA developed less severe infection when inoculated with the pathogen. Also, infected plants that were treated at 24h, 6 days, and 14 days after infection were healthier on average six weeks later. Photography, microscopy, and physiological measurements were used to support the treatment claims.

Overall, this manuscript has potential to make a good contribution to the literature. The findings are novel in that dsRNA has not been used extensively to treat fungal disease after visible symptoms begin. The chosen dsRNAs appear efficacious; the chosen metrics are relevant for plant disease; the manuscript is written clearly; and the images and micrographs are clearly presented. However, I have some concerns. My main issue is that treatment effect sizes are not sufficiently documented. For example, “prevention” as in Fig. 1 is a claim of clinically meaningful significance; statistical significance is only one part of that. Please estimate the effect size in the form of diseased area reduction, fewer lesions, smaller lesions, or some other relevant metric with units, and present it in the text.

I also take issue with the frequent use of “cure” and “curative.” My (perhaps uninformed) understanding of “cure” is more or less complete cessation of disease symptoms, ideally accompanied by a negative test for the pathogen itself. This is the common usage of the term (e.g. “penicillin cures some types of bacterial infection”). At present it seems that the words “treatment” or “inhibition” are more warranted. In Fig. 3 some plants may indeed be cured based on this stringent definition, although the effect is noisy and given the small sample size, 2-3 out of 6 plants look similar to controls. Here too it would strengthen the authors’ claims to discuss effect sizes. Finally, some claims in the introduction do not seem to match what is shown in the results.

I support the publication of this manuscript provided that the authors revise and clarify some points. Line-by-line comments, questions, and suggestions are as follows.

- 1. (From previous text) Please estimate the effect size in the form of diseased area reduction, fewer lesions, smaller lesions, or some other relevant metric with units, and present it in the text.*

Thank you for this recommendation to consider effect size. As you pointed out, this is a meaningful metric that further strengthens our preventative and curative data. We have now quantified effect size, with respect to disease area and plant health in all preventative and curative assays using Hedge’s *g*, which measures the standardized difference between two means (as Cohen’s *d* does) but also corrects

for potential bias in smaller sample sizes. These data have been presented in text at lines 100-103 (preventative), lines 165–167 & 171-172 (24 hours curative), lines 189–191 & 195–198 (6 days curative), and lines 224–226 (14 days curative). Effect sizes are mentioned as a measure for disease and plant health in the discussion at lines 274–275. The computational methods for Hedge’s g have also been included at lines 469–471 (Supplementary table 1). The raw data is attached below:

Supplementary Table 1. Effect sizes of disease coverage and plant health measures in control and treatment groups at all preventative and curative timepoints. Effect sizes are calculated according to Hedge’s g, and effect sizes of 0.15, 0.40, and 0.75 are considered as small, medium, and large, respectively.

Disease Coverage								
Preventative			Curative 24 hr			Curative 6 dpi		
Group 1	Group 2	Effect size	Group 1	Group 2	Effect size	Group 1	Group 2	Effect size
Control	GFP	0.39	Control	GFP	0.23	Control	GFP	0.10
Control	BTUB	3.38	Control	BTUB	2.64	Control	BTUB	4.12
Control	EF1-A	3.81	Control	EF1-A	2.38	Control	EF1-A	3.29
GFP	BTUB	3.44	GFP	BTUB	2.90	GFP	BTUB	2.98
GFP	EF1-A	3.44	GFP	EF1-A	2.65	GFP	EF1-A	2.48
BTUB	EF1-A	0.21	BTUB	EF1-A	0.12	BTUB	EF1-A	0.86
Plant Health								
Curative 24 hr			Curative 6 dpi			Curative 14 dpi		
Group 1	Group2	Effect size	Group 1	Group 2	Effect size	Group 1	Group 2	Effect size
Control	GFP	0.19	Control	GFP	0.13	Control	GFP	0.51
Control	BTUB	1.84	Control	BTUB	2.81	Control	BTUB	1.68
Control	EF1-A	2.21	Control	EF1-A	1.78	Control	EF1-A	2.54
GFP	BTUB	1.64	GFP	BTUB	2.27	GFP	BTUB	1.55
GFP	EF1-A	2.02	GFP	EF1-A	1.55	GFP	EF1-A	2.46
BTUB	EF1-A	0.40	BTUB	EF1-A	0.56	BTUB	EF1-A	0.33

2. *(From previous text) I also take issue with the frequent use of “cure” and “curative.” My (perhaps uninformed) understanding of “cure” is more or less complete cessation of disease symptoms, ideally accompanied by a negative test for the pathogen itself. This is the common usage of the term (e.g. “penicillin cures some types of bacterial infection”). At present it seems that the words “treatment” or “inhibition” are more warranted.*

We adopted the terms ‘cure’ and ‘curative effect’ from the most applicable existing study, which demonstrates the ‘curative’ effect of dsRNA against an oomycete (a fungal-like organism) causing downy mildew on Grapevine (Haile *et al.* 2021). We believe that in this field this is the most appropriate language to explain the significantly reduced disease symptoms and restoration of plant health we saw with the application of dsRNA following establishment of disease.

We can appreciate how the use of the word ‘cure’ could be misleading, given the classic clinical definition, and may be different to the terms curative treatment/curative role/curative effect. In light of your concerns and suggestions, we have included a study-specific definition of ‘prevent’ and ‘cure’ on lines 72-78.

This now reads as “Rust fungi are obligate pathogens. While some taxa occur as latent infections, this has not been demonstrated for *A. psidii*, in which the presence of diseased vs healthy leaves is adequate to determine whether the fungus is present on the host³⁵. In this study, the terms ‘cure’ or ‘curative’ denote the application of dsRNA after *A. psidii* infection resulting in significantly reduced disease symptoms and a restoration of plant health. ‘Prevent’ or ‘preventative’ signifies dsRNA application before infection leading to significant prevention or inhibition of myrtle rust symptoms.”

3. *48-49 Does forest management practice currently include/allow foliar fungicide treatment? Introduction and Discussion both mention that dsRNA could be applied in a wild setting, so it would be beneficial to have sentence describing current in situ conservation efforts.*

Yes, in Australia, triadimenol fungicide is registered for the control of *A. psidii* (<http://permits.apvma.gov.au/PER12319.PDF>). Carnegie *et al.* (2016) found that spray-application of triadimenol to *Rhodamnia rubescens* and *Rhodamnia psidiodes* (highly susceptible hosts of myrtle rust) was temporarily effective in suppressing myrtle rust symptoms, but that trees would re-infect after fungicide applications stopped. This information has now been added to the introduction at lines 45–47.

4. *68-70. “cure plants from severe fungal infections and prevent further infections” Please clarify. “Further” implies that the plants were first cured of infection, and then became more resistant to subsequent pathogen challenge. This study did not test that, and it is a different*

goal than than the aims to “(i) prevent infection, and ii) cure established infections” (line 70). Suggestion: introduce the aims in the order they are dealt with in the results.

We agree that this word choice was confusing and we have now modified it to “Here, we used myrtle rust as a model system to build on our previous study, showing that dsRNA co-inoculated with urediniospores inhibited infection physiology of two rust fungi²⁶ – to determine whether dsRNA could prevent future infections, as well as cure plants from severe existing infections” so as to be consistent with the paper’s aims and order of results (lines 70–72).

5. 69 *“Our overall aim was to improve plant performance and conservation outcomes” Doesn’t seem accurate; conservation outcomes were not tested in this study. Suggestion: “Our overall aim was to improve measures of plant health with the eventual goal of application in conservation efforts.”*

Upon reflection we agree that this wording was not quite accurate, and have amended this sentence in line with your suggestion. This now reads as “Our overall aim was to improve plant health and recovery, following infection by *A. psidii*, with the eventual goal of application in conservation efforts” (lines 79–80).

6. 86 *Clear images with an obvious treatment effect. It would be nice to see a flat leaf image representing each treatment group. I assume the “Leaf Doctor” app takes flat images as the input, correct? This could be a supplemental.*

As per your recommendation, we have produced an additional supplementary figure that includes a representative leaf image from each treatment group, at each time point where percent disease coverage was assessed with the Leaf Doctor application (Supplementary Figure 1).

Supplementary Figure 1. Representative leaves used for disease assessments in the Leaf Doctor application from *Syzygium jambos* plants infected with *Austropuccinia psidii* at all preventative and curative timepoints. *Syzygium jambos* trees (n=6) were infected with *A. psidii* urediniospore inocula and treated with a nuclease free H₂O (negative control), a non-specific dsRNA control (*GFP* dsRNA), or *beta-tubulin* (β -*TUB*), or *transcription elongation factor* (*EF1- α*) *A. psidii*-specific dsRNAs at a concentration of 100 ng/ μ L at 48 hours pre-infection, 24 hours, or 6 days, post-infection. Each leaf is representative of technical and biological replicates in its treatment or control group. In the Leaf Doctor application, areas that are yellow, orange, pink, or red in colour will be scanned as ‘diseased’ and areas that are various shades of green will be scanned as ‘healthy’, allowing the application to determine a percent (%) disease coverage of leaf.

Actioned

8. 98 *asterisks

Actioned

9. 103 Did you try combining the two dsRNAs (EF1-a and b-TUB) 1:1 as a combination therapy? Not required but I'm curious if there might be synergy.

We have never formally tested a combination treatment of the two dsRNAs. Out of the same curiosity, we have done some preliminary experiments combining both with *in vitro* germinating *A. psidii* urediniospores and saw a very similar result to what is seen in individual treatments. However, we have never formally quantified these results and did not proceed with these experiments as the treatments are highly effective when administered individually, and the impacts on *A. psidii* infection structure morphology were very similar (as shown in our previously published work), indicating that the treatments are unlikely to further complement one another.

10. 123 This sentence is also in Methods. It's fine if it appears only in Methods.

This sentence has now been deleted from the figure caption and appears only in the methods.

11. 125-126 “dsRNA targeting *A. psidii* can cure an established infection” Suggestion: “dsRNA targeting *A. psidii* can effectively treat an established infection.” The treatment strongly inhibits disease progression, but in my opinion this does not warrant the word cure without more explanation.

We have now provided further explanation and clarification of terminology at lines 72–78.

12. 198 *asterisks

Actioned.

13. 201-202 Standardize abbreviation for stomatal conductance. Sometimes it is lowercase with subscript and other times not.

This has now been standardized to g(subscript)sw.

14. 209 *asterisks

Actioned.

15. 264 “positive flow-on effect” Word choice — not sure what is meant here. Suggestions: knock-on effect / positive feedback / virtuous circle. Maybe there is a standard epidemiology term for this kind of effect?

This now reads as: “Curative treatments may provide further relief, preventing or lowering production of viable new inoculum, and inhibiting epidemic disease cycles” (line 324-325).

16. 291 *If you bring up the practicality of dsRNA application, it would help to compare it to current practices, if any. 100 ng RNA / uL = 100 mg / L, correct? Can this be produced at a scale sufficient for use in the field or greenhouse? How does this compare to the cost of a conventional foliar fungicide? These questions are optional and beyond the scope of the current work, but could help to get people inspired if you advocate for more widespread dsRNA use.*

Recent research efforts to produce dsRNA rapidly and in a cost-effective way through large-scale microbial-based pipelines have led to a dramatic reduction in cost of Technical Grade Active Ingredient (TGAI) dsRNA. Costs per gram of TGAI dsRNA decreased from \$12,500 USD in 2008, to \$1 USD in 2023 (de Andrade and Hunter, 2016; Zotti et al., 2018; Dalakouras et al., 2020; GreenLight, 2023).

In addition, the time from ideation to commercial launch can also be significantly faster (45%) for RNA-based biopesticides compared to conventional agrichemicals (GreenLight, 2023).

An overview of this has now been included in the discussion from lines 352–353. This reads as “Additionally, large-scale microbial-based dsRNA production pipelines now allow for cost-effective synthesis of dsRNA, down from \$12,500 USD per gram in 2008, to as little as \$1 USD per gram in 2023^{55–57}”.

17. 295 *“harmful fungicides in industry” If you bring up toxicity of conventional fungicides, you also need to mention potential off-target effects of dsRNA on benign and beneficial species. The target genes appear very conserved, and the text does not mention any effort to make the dsRNA sequences specific to A. psidii.*

We have added a sentence to this paragraph describing the specificity of the dsRNA to the *A. psidii* genome (which prevents off-target effects) and have referenced back to our previous publication, which explains in detail the design of these constructs.

This reads as “This treatment option does not need a carrier to protect dsRNA when applied to plants that are already infected, and dsRNAs are designed to specifically target *A. psidii* barcoding genes²⁶, so as to prevent any off-target effects” (lines 349–352). In addition, the target sequences of dsRNA constructs used in this publication and in our previous publication, are designed off barcoding genes, so are unlikely to result in off-target effects as these genes contain regions which are specifically used to identify species.

18. 305 *Please provide a citation number. I assume this is Ref. 25.*

Actioned

19. 328-330 Please provide more detail on area measurements. Fig. 1 legend states that this is “mean (n = 6) percent coverage” of new growth leaves, and that there are 6-8 leaves per tree. Methods should specify how these measurements were aggregated. Were individual leaf measurements pooled for the leaves on a particular tree, which then became an individual dot on the box plot?

Further details have now been included explaining technical replicates (leaves) and pooling of technical replicates into data points. This now reads as “Six biological replicates (trees) were used per treatment and 6 – 8 technical replicates (leaves) were measured per tree, depending on the number of new leaves susceptible to infection. Technical replicates from each tree were aggregated into to one data point (biological replicate) for statistical analysis and in figures” (lines 402–404).

REVIEWERS' COMMENTS:

Reviewer #1 (Remarks to the Author):

The authors have addressed most of my comments or concerns in the revised manuscript. I recommend it to be accepted as it is.

Reviewer #3 (Remarks to the Author):

The authors addressed my concerns to my satisfaction. I appreciate the addition of effect sizes and flat leaf images. I am OK with use of the word "curative" as long as this is defined clearly in the introduction and there are citations for previous work establishing the use of the word in this way. The manuscript is looking good.